# Thirty-Year Urbanization Trajectories and Obesity in Modernizing China

**DOI:** 10.3390/ijerph19041943

**Published:** 2022-02-09

**Authors:** Wenwen Du, Huijun Wang, Chang Su, Xiaofang Jia, Bing Zhang

**Affiliations:** National Institute for Nutrition and Health, Chinese Center for Disease Control and Prevention, Beijing 100050, China; duww@ninh.chinacdc.cn (W.D.); wanghj@ninh.chinacdc.cn (H.W.); suchang@ninh.chinacdc.cn (C.S.); jiaxf@ninh.chinacdc.cn (X.J.)

**Keywords:** urbanization, trajectory, group-based trajectory modeling, obesity, BMI, weight

## Abstract

The effects of long-term urbanization changes in obesity are unclear. Data were obtained from the China Health and Nutrition Survey (CHNS) 1989–2018. A multidimensional urbanicity index was used to define the urbanization level for communities. Group-based trajectory modeling was used to identify distinct urbanization change trajectories. Gender-stratified multilevel models were used to investigate the association between urbanization trajectories and weight/BMI, through the PROC MIXED procedure, as well as the risk of being overweight + obesity (OO)/obesity (OB), through the PROC GLIMMIX procedure. A total of three patterns of the trajectory of change in urbanization were identified in 304 communities (with 1862 measurements). A total of 25.8% of communities had a low initial urbanization level and continuous increase (termed “LU”), 22.2% of communities had a low–middle initial urbanization level and constant increase (termed “LMU”), and 52.0% of communities had a middle–high initial urbanization and significant increase before 2009, followed by a stable platform since then (termed “MHU”). During the 30 follow-up years, a total of 69490 visits, contributed by 16768 adult participants, were included in the analysis. In the period, weight and BMI were observed in an increasing trend in all urbanization trajectory groups, among both men and women. Compared with LU, men living in MHU were related to higher weight, BMI, and an increased risk of OO (OR: 1.46, 95%CI: 1.26 to 1.69). No significant associations were found between urbanization trajectories and OB risk in men. Among women, the associations between urbanization and all obesity indicators became insignificant after controlling the covariates. Obesity indicators increased along with urbanization in the past thirty years in China. However, the differences among urbanization trajectories narrowed over time. More urbanized features were only significantly associated with a higher risk of obesity indicators in Chinese men. The effects of urbanization on obesity among women were buffered.

## 1. Introduction

Since the population residing in urban areas is increasing rapidly worldwide, the influence of urban features on health has greater importance [1,2]. Urbanization is defined as urban population size or density and usually explores the association of health by an urban–rural dichotomy [3,4]. This simple classification, thereby, misses sectors that are in an urban environment and have a potential influence on health, such as infrastructure, social service, communications, and housing. In recent years, multi-dimension indexes were used to indicate the extent of urbanization [5,6], which may provide broader insights.

Obesity was considered a result of modern environments, coupled with inactive lifestyles and energy-dense diets, and became a growing public health challenge in both China and the world [7,8,9]. With rapid urbanization in the past three decades, since the 1990s, obesity has increased tremendously in China [10].

Studies exploring the association between urbanization and obesity provided mixed results. Unlike other developing countries, where obesity increased more in urban areas [11,12], China showed higher obesity prevalence and a faster increase in rural versus urban areas for years [1,13]. It could be hypothesized that the population in less or more urbanized areas may have progressed in nutrition and lifestyle transitions at different rates [14,15]. Due to the rapid development of the economy since the 1990s, China provided a unique opportunity to explore the association between urbanization and obesity.

However, the bulk of the literature was based on the stable classification of urbanization [16,17,18] and was, thus, unable to obtain the association between dynamic changes in urban features and obesity. Our study extended the previous findings on the association of urbanization and obesity in the following aspects. We used a continuous, validated urbanicity index that captured 12 dimensions of urban features [19], which may provide more nuanced aspects of urbanization than the simple rural–urban dichotomy. Furthermore, instead of a fixed definition of urbanization, we used group-based trajectory modeling to identify distinct groups with similar underlying trajectories of urbanicity index [20].

To provide scientific evidence for the association between long-term changes in urbanization and obesity outcomes, we used thirty years of data from a large nationwide cohort study, the China Health and Nutrition Survey (1989–2018). We hypothesized that populations residing in communities with different urbanization trajectories may have variable risks of obesity. Gender disparities were also explored in this paper to examine whether men and women cope with urbanization differently.

## 2. Materials and Methods

### 2.1. Data

We used data from the China Health and Nutrition Survey (CHNS) (1989–2018), an ongoing longitudinal cohort study, which aims to explore the influences of social changes on population health in China. Eleven waves of surveys were sequentially conducted in 1989, 1991, 1993, 1997, 2000, 2004, 2006, 2009, 2011, 2015, and 2018. The coverage increased from 8 provinces in 1989 to 16 provinces in 2018. The CHNS used a multistage, random cluster sampling method to select 24 communities from each province. Then, 20 households were selected in each community. All individuals in the households were interviewed in the survey. Detailed procedures and the study design were published elsewhere [21,22].

### 2.2. Sample Selection

To describe urbanization trajectories, we confined the trajectory analysis to 304 communities with urbanization measures in three and more waves of surveys. The individual analysis sample was limited to adults aged 18 to 64 years old, having at least two survey visits with complete height and weight measurements, excluding pregnant women (*n* = 518 visits). The size of analytic participants by number of repeat visits was presented (two visits, *n* = 5421; three visits, *n* = 3472; four visits, *n* = 2024; five visits, *n* = 1556; six visits, *n* = 1288; seven visits, *n* = 1054; eight visits, *n* = 903; nine visits, *n* = 607; ten visits, *n* = 310; and eleven visits, *n* = 133; total *n* = 16,768 participants, across 69,490 visits).

### 2.3. Urbanization Trajectories

A validated 12-component urbanization index was used to capture the degree of urbanization in the communities. The index was developed specifically for the CHNS, combining individual- and community-level data to represent multicomponent domains that could distinguish the urban features in China. The scale included the following components: (1) population density; (2) economic activities; (3) traditional markets; (4) modern markets; (5) transportation; (6) sanitation; (7) communications; (8) housing; (9) education; (10) community diversities in education and income level; (11) health infrastructure; and (12) social services. The points were allocated in each of the above 12 domains, with a possible total range of 0–120. The higher score reflected more urban features across the multiple domains.

In the present study, patterns of change in urbanization were identified using group-based trajectory modeling, a method that has recently been used to identify distinct groups with similar trajectories in epidemiological data [23,24]. Group-based modeling was conducted using SAS version 9.4 (SAS Institute, Cary, NC, USA), with the TRAJ procedure. The following criteria were used to determine the best fit models: (1) lowest Bayesian information criterion and (2) inclusion of at least 5% of the sample size. Once the urbanization trajectories were determined, the communities were assigned to the class with the highest posterior probability. Trajectory membership was then used as an indicator variable in the main analyses.

### 2.4. Dependent Variables

Height and weight were measured by trained physicians and nurses in each survey, according to the same protocols. Height, without shoes, to the nearest 0.1 cm, was measured by a portable SECA stadiometer (SECA, Hamburg, Germany). Body weight was measured, without shoes and with light clothing, to the nearest 0.1 kg, by a calibrated beam scale.

To examine the impacts of urbanization trajectories on obesity, we used weight (kg) and BMI (kg/m^2^) as the continuous dependent variables, as well as overweight + obesity (OO, BMI ≥ 24 kg/m^2^) and obesity (OB, BMI ≥ 28 kg/m^2^) as the categorical dependent variables [25].

### 2.5. Covariates

We included the following sociodemographic and behavior factors as covariates in our analyses: survey year, age (18–29, 30–44, and 45–64 years) [26], gender (men and women), education level (primary school or less, junior high school, senior high school, and postsecondary), current smoking (yes or no), alcohol drinking in the past year (yes or no), household income per capita (inflated to 2018 and categorized into tertiles), and entry weight (kg)/BMI (kg/m^2^). The daily energy intake was estimated by combination of household inventory weighing and individual 24-h dietary recall for three consecutive days and then was categorized into tertiles (low, medium, and high). Furthermore, we derived weekly physical activity (PA) measures from a detailed seven-day PA recall assessment to calculate the metabolic equivalent of task hours per week and then categorized them into tertiles (low, medium, and high). All of the covariates included in our analyses were time-varying, except for gender.

### 2.6. Statistical Analysis

The identification of urbanization trajectories was through the SAS PROC TRAJ procedure, using the censored normal model. We described demographic characteristics of entry sample by gender using mean ± sd for continuous variables and proportion for binary and categorical variables. Predicted weight/BMI by urbanization trajectories in men and women were generated from multilevel mixed models, respectively, controlled for survey year, education level, smoke, drinking, age, income, dietary energy intake, physical activity, entry age, and entry weight/BMI.

Gender-stratified multilevel mixed-effect regression models were then used to investigate the association between urbanization trajectories and weight/BMI through the PROC MIXED procedure in SAS, as well as the risk of being OO/OB through the PROC GLIMMIX procedure. We used steps for adjustment; the crude model (Model 1) only included urbanization trajectories (ref = ”LU” trajectory). Then, two adjusted models were built: (1) Model 2 was additionally adjusted for socioemographic, lifestyle, and dietary covariates, including survey year, education level, smoke, drinking, age, income, dietary energy intake, and physical activity, based on Model 1; (2) Model 3 was further adjusted for initial covariates, including entry age and entry weight/BMI, based on Model 2.

All statistical analyses were performed using SAS 9.4 (SAS Institute, Cary, NC, USA). Two-tailed *p* < 0.05 was considered statistically significant.

## 3. Results

### 3.1. Urbanization Trajectories

Among 304 communities, with 1862 urbanization measurements, we identified three distinct trajectories of change in the urbanicity index (Figure 1), according to the highest posterior probability. The first trajectory (labeled “1”), characterized by a low initial urbanization level and continuous increase during the follow-up period, was termed “LU”, which represented 25.8% of communities. The second trajectory (labeled “3”), characterized by a low–middle initial urbanization level and constant increase, was termed “LMU”, which represented 22.2% of communities. The last trajectory (labeled “2”) was characterized by a middle–high initial urbanization and significant increase before 2009, followed by a stable platform since then, was termed “MHU”, which represented 52.0% of communities.

### 3.2. Characteristics of Entry Participants

The analytic individuals were 16,768 persons, including 8094 men and 8674 women (Table 1). The mean age of participants at the baseline survey was 36.99 ± 12.31 years. The initial weight and BMI were 58.73 ± 10.72 kg and 22.4 ± 3.23 kg/m^2^, respectively. Almost 77% of the participants were enrolled in the first survey before 2009, and only 22.66% of individuals participated in the first survey after 2009. About 27.11%, 24.78%, and 48.12% of participants lived in LU, LMU, and MHU communities, respectively. In the entry sample, 26.93% of participants were defined as OO, with a BMI ≥ 24 kg/m^2^, and 5.81% were defined as OB, with a BMI ≥ 28 kg/m^2^.

### 3.3. Association between Urbanization Trajectories and Weight and BMI by Genders

Table 2 shows the results of the multilevel mixed regression models, investigating the association between urbanization trajectories and weight/BMI in men and women separately. For men, only the MHU trajectories were found to be associated with higher weight, compared to LU after adjusting for potential covariates, including entry age and entry weight. LMU trajectory was not significantly associated with weight in men. Similar findings were found for BMI in men. According to the results, the weight and BMI of men living in MHU communities was 0.61 kg and 0.19 kg/m^2^ higher than those living in LU trajectory communities (*p* < 0.05). For women, after controlling for all the covariates, no significant associations with weight/BMI were shown in both LMU and MHU trajectory patterns.

To facilitate the interpretation of the model findings, we predicted weight (Figure 2a,b) and BMI values (Figure 2c,d) by urbanization trajectories in men and women. We observed that the predicted weight and BMI increased during the past decades in all urbanization trajectory groups among Chinese adults. However, the differences seemed to be unclear, especially for women. We found that men living in MHU communities were more likely to have higher weight and BMI during the overall follow-up years (*p* < 0.05). The predicted weight/BMI in men showed no significant differences between LMU and LU communities. Furthermore, we also observed no significant differences in urbanization trajectories in women. An interesting point is that, since 2011, women living in LMU communities have seemed to have a higher BMI than those living in LU and MHU areas.

### 3.4. Association between Urbanization Trajectories and Overweight + Obesity and Obesity by Genders

Considering the potential covariates, the multilevel mixed regression results (Table 3) showed that, compared with LU trajectory, MHU trajectory was associated with a higher risk of OO in men (OR: 1.46, 95%CI: 1.26 to 1.69, *p* < 0.05). No significant associations were shown between urbanization trajectories and OB in men (*p* > 0.05). In women, the associations between urbanization trajectories with neither OO nor OB were significant (*p* > 0.05).

## 4. Discussion

### 4.1. Main Findings

China has experienced rapid economic growth since the 1980s. In the current study, we investigated the urbanization change in Chinese communities using group-based trajectory modeling, based on a thirty-year longitudinal study. Three patterns of urbanization trajectories were identified. During the past decades, rapid urbanization was observed in Chinese communities, no matter whether the initial urbanization level was low, middle, or high. It is worth noting that the “MHU trajectory”, which contributed to about half of the community sample, had a rapid increase in urbanization before 2009 and kept stable in the following years. Meanwhile, the “LU trajectory” and “LMU trajectory” communities also had a significant increase in urbanization and, finally, reached the initial level of “LMU trajectory” and “MHU trajectory”, respectively. This implies that nutrition and lifestyle changes, as well as the increasing chronic diseases burden, accompanied with urbanization, will be observed in less urbanized areas in the future.

We hypothesized that the role of urbanization in the burden of obesity was patterned by gender. The results we found in the present study suggested that living in communities with more urban features was positively related to higher weight and BMI, as well as an increased risk of OO and OB. However, the associations observed in women were attenuated after adjusting for the initial weight/BMI and baseline age. It was still observed that living in MHU communities significantly increased the risk of OO in Chinese men, even after adjusting for baseline age and BMI.

### 4.2. Comparison with Previous Studies

In developed countries, obesity prevalence was reported as higher in less urbanized communities than in more urbanized communities [27,28]. A similar trend was also observed in some developing countries in recent years [29,30]. It was shown that more than 80% of the rise in BMI in some low- and middle-income regions was due to increases in rural areas, which became the main driver of the global obesity epidemic [31]. Increasing urbanicity, which defined according to a multi-component scale, was found to be associated with a high body mass index in rural communities in Uganda [32]. Obesity and related chronic diseases have increased as one of the major health burdens shifting towards the poor [33]. The general hypothesis of increasing obesity in developing countries was the rapid transition in physical activity and diet behaviors, as well as the overall food system with urbanization [34,35,36].

Urbanization is a dramatic process and may result in divergent influences on health at different stages. Increasing urbanization was found to reduce carbohydrate consumption consistently and reduce fat consumption after a turning point [37]. Abdominal adiposity in China was found to be positively associated with urbanization in the 1990s and changed to a negative association in the 2000s [1]. Another study indicated that the association of urbanization with hypertension was also not linear, which meant higher urbanization, to some extent, might turn out to be a protective factor [2]. Changes in early urabanizing stage on community food environment, such as increasing availability of restaurants and supermarkts, as well as community norms for fast food consumption [38], may be one of the explanations for the obesity challenge. However, the effect of urbanization on health varied by social and geographic space across gender.

With the “urban–rural coordination development” strategy, proposed by the Chinese central government, rural residents’ living in China improved dramatically [37,39]. They also experienced the rapid transition towards energy-dense diets and lower physical activities, which implied a narrowing gap in the risk of obesity with more urbanized areas [40,41,42,43]. Influences of urbanization on food consumption were found to be both indirect and multidimensional [44]. Evidence of income buffering the transition from traditional to modern diets could be one of the potential explanations for limited urban–rural differences in Western diet behaviors at the later stages of urbanization [45].

Unlike the results found in the Western population [27] (in which prevalence of obesity was only significantly greater among women living in nonmetropolitan statistical areas, compared with women living in large metropolitan statistical areas, while the prevalence of severe obesity in nonmetropolitan statistical areas was higher than in large metropolitan statistical areas among both men and women), in the current study, we obtained a higher risk of OO among men living in areas with more urbanized features than those in less urbanized areas, while there were no significant differences by urbanization trajectories among women. The consecutive nationally reprehensive surveys in China also indicated that the mean BMI in 2018 was already higher in rural than urban women but still remained lower in rural men [8].

Given that the association between urbanization and risk of OO remained statistically significant in men, after adjusting for demographic-, nutrition-, and lifestyle-related factors, it may stem from the hypothesis that other aspects of more urbanization environments may be associated with an increased risk of obesity among men. However, we could not conclude similar findings for women, as the positive relationship in women shifted to insignificant, after adjusting the initial obesity indicators, which implied that women might take potential behavior modifications during the urbanized period.

### 4.3. Strengths and Limitations

There are some limitations to this study. First, we used the urbanicity index to demonstrate the urbanization level in the current study, which was specifically developed and validated in the China Health and Nutrition Survey and, thus, could not be directly applied in other studies and compared with similar results obtained worldwide. Additionally, the index cannot capture the current rapid transition and needs to be updated in the future. Second, limited research on rapid urbanization and obesity makes these findings difficult to compare with other studies.

Despite the above limitations, this study extends the previous knowledge on urbanization and obesity in developing countries. In many ways, the detailed thirty-year CHNS, with eleven repeated measures, provides a remarkable window through which to understand urbanization-related changes in other low- and middle-income countries (LMICs). Specifically, we observed the long-term trajectories of urbanization in China and explored the different associations with obesity indicators across gender. It is meaningful for policymakers to develop specific and effective initiatives for obesity issues, accompanied by rapid urbanization, in different populations.

Authors should discuss the results and how they can be interpreted from the perspective of previous studies and the working hypotheses. The findings and their implications should be discussed in the broadest context possible. Future research directions may also be highlighted.

## 5. Conclusions

China has experienced rapid urbanization in the past 30 years. Our results indicate gender disparities of obesity risk in the urbanization period. Living in MHU communities was associated with higher weight/BMI and increased odds of OO among Chinese men. The present study adds to our understanding that higher urbanized features may be more challenging for healthy weight among men than women in China. On the one hand, the rural population, especially rural women, needs more attention for their rapid increase in obesity indicators at a similar, or even faster rate, than women in urban areas. Other obesity-related chronic diseases should be explored, in relation to urbanization, in the future to address more health risks and vulnerable populations in modernizing China.

## Figures and Tables

**Figure 1 ijerph-19-01943-f001:**
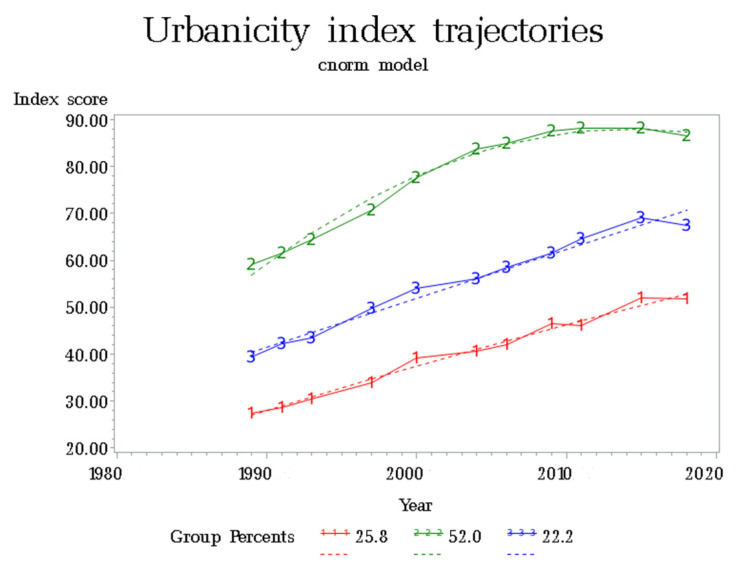
Trajectory modeling identified three urbanization patterns.

**Figure 2 ijerph-19-01943-f002:**
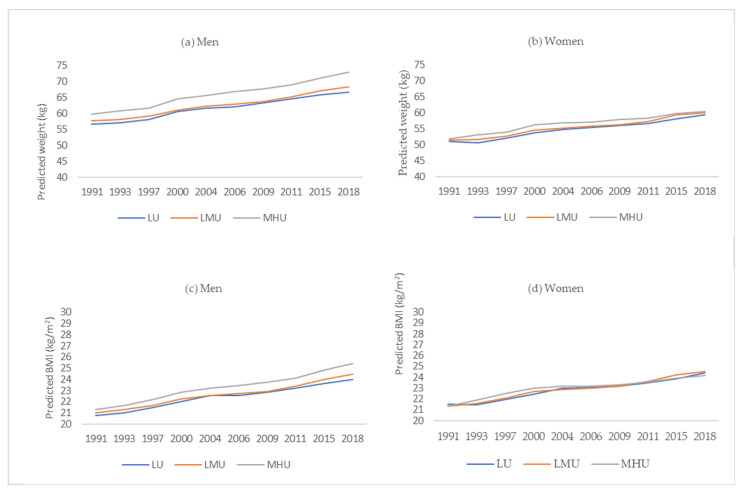
Predicted weight in men (**a**) and women (**b**) and predicted BMI in men (**c**) and women (**d**) by urbanization trajectories. Predicted weight/BMI was generated from multilevel mixed models, controlled for survey year, education level, smoke, drinking, age, income, dietary energy intake, physical activity, entry age, and entry weight/BMI. Abbreviations: BMI, body mass index; LU, low urbanization; LMU, low–middle urbanization; MHU, middle–high urbanization.

**Table 1 ijerph-19-01943-t001:** Demographic characteristics of the baseline sample by gender.

	Men	Women	Overall
N	8094	8674	16,768
Age (years)	36.79 ± 12.63	37.17 ± 12	36.99 ± 12.31
Weight (kg)	62.69 ± 10.85	55.03 ± 9.17	58.73 ± 10.72
BMI (kg/m^2^)	22.33 ± 3.18	22.46 ± 3.28	22.4 ± 3.23
Entry year (%)			
1989	26.77	27.07	26.93
1991	18.16	16.46	17.28
1993	3.98	2.7	3.32
1997	12.64	11.53	12.06
2000	7.34	7.98	7.67
2004	6.6	7.02	6.82
2006	3.32	3.2	3.26
2009	5.91	6.41	6.17
2011	10.75	11.72	11.25
2015	4.53	5.9	5.24
Urbanization trajectories (%)			
LU	28.07	26.2	27.11
LMU	25.17	24.42	24.78
MHU	46.76	49.38	48.12
^a^ OO (%)	25.91	27.88	26.93
^b^ OB (%)	5.76	5.86	5.81

^a^ OO was defined as BMI ≥ 24 kg/m^2^; ^b^ OB was defined as BMI ≥ 28 kg/m^2^. The continuous variables (age, weight, and BMI) were expressed as mean ± sd.; abbreviations: BMI, body mass index; LU, low urbanization; LMU, low–middle urbanization; MHU, middle–high urbanization; OO, overweight; OB, obesity.

**Table 2 ijerph-19-01943-t002:** Multilevel mixed regression estimates, examining weight and BMI by gender.

Urbanization Trajectories	Weight			BMI		
	Model 1	Model 2	Model 3 ^a^	Model 1	Model 2	Model 3 ^b^
Men						
LU (ref)	0	0	0	0	0	0
LMU	1.64 *	1.11 *	0.19	0.44 *	0.30 *	0.10
MHU	5.52 *	4.17 *	0.61 *	1.17 *	0.81 *	0.19 *
Women						
LU (ref)	0	0	0	0	0	0
LMU	1.27 *	1.05 *	0.27	0.30 *	0.29 *	0.11
MHU	2.80 *	2.31 *	0.21	0.45 *	0.47 *	0.03

Model 1 includes urbanization trajectories only; Model 2 includes survey year, education level, smoke, drinking, age, income, dietary energy intake, and physical activity, based on Model 1; Model 3 ^a^ includes entry age and entry weight on Model 2; Model 3 ^b^ includes entry age and entry BMI on Model 2; *, *p* < 0.05; abbreviations: BMI, body mass index; LU, low urbanization; LMU, low–middle urbanization; MHU, middle–high urbanization.

**Table 3 ijerph-19-01943-t003:** Multilevel mixed regression analysis of associations between urbanization trajectories and risk of obesity by gender.

Urbanization Trajectories	OO (BMI ≥ 24 kg/m^2^)			OB (BMI ≥ 28 kg/m^2^)		
	Model 1	Model 2	Model 3	Model 1	Model 2	Model 3
Men						
LU (ref)	1	1	1	1	1	1
LMU	1.35(1.18,1.55) *	1.22(1.05,1.42) *	1.14(0.98,1.31)	1.56(1.29,1.89) *	1.36(1.11,1.66) *	1.22(0.99,1.52)
MHU	2.53(2.24,2.85) *	1.97(1.70,2.27) *	1.46(1.26,1.69) *	2.13(1.80,2.52) *	1.60(1.31,1.96) *	1.09(0.87,1.35)
Women						
LU (ref)	1	1		1	1	1
LMU	1.15(1.00,1.32) *	1.17(1.01,1.35) *	1.07(0.93,1.22)	1.21(1.01,1.45) *	1.24(1.02,1.50) *	1.05(0.85,1.29)
MHU	1.35(1.20,1.52) *	1.44(1.24,1.66) *	1.10(0.95,1.27)	1.35(1.15,1.58) *	1.49(1.23,1.81) *	1.03(0.83,1.27)

Model 1 only includes urbanization trajectories; Model 2 includes survey year, education level, smoke, drinking, age, income, dietary energy intake, and physical activity, based on Model 1; Model 3 includes entry age and entry BMI on Model 2; *, *p* < 0.05.

## Data Availability

Not applicable.

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
