# Peer review of "Thirty-Year Urbanization Trajectories and Obesity in Modernizing China"

_ijerph, 2022, doi:10.3390/ijerph19041943_

Round 1

Reviewer 1 Report

  1. Body weight is actual body weight or ideal body weight?
  2. Figure 1; the title "index traj" is not professional; please do not use the informal abbreviations in figures
  3. Figure 1; how to interpret X- and Y- axis should be provided in the figure 
  4. Table 1 and Table 2, will need to have all abbreviations below the table. 
  5.   Figure 2, will need to have all abbreviations below. Also, will need to describe what different colors mean
  6.  In method, will need to provide more details how to do multivariate adjusted models  that are shown in Table 3.
  7.  Clinical implications of this study and future directions are needed to imrpove

Author Response

Thank you for the invaluable comments that have enabled us to improve this manuscript. The manuscript has been revised according to the comments and journal style. Below is a detailed response:

1. Body weight is actual body weight or ideal body weight?

Response: Thank you for this comment. We used actual body weight to build the models.

2. Figure 1; the title "index traj" is not professional; please do not use the informal abbreviations in figures

Response: Thank you for the suggestions. The title was modified to “Urbanicity index trajectories”.

3. Figure 1; how to interpret X- and Y- axis should be provided in the figure 

Response: Thank you for the suggestions. X- and Y-axis title were added as “Year” and “Index score”, respectively.

4. Table 1 and Table 2, will need to have all abbreviations below the table. 

Response: Thank you for this comment. Abbreviations were added below the tables, as well as clarified at the first place mentioned in the manuscript.

5. Figure 2, will need to have all abbreviations below. Also, will need to describe what different colors mean

Response: Thank you for the valuble comments. Abbreviations were added below the figure, as well as clarified at the first place mentioned in the manuscript. Different colors mean the three urbanization trajectories derived from group-based trajectory modeling, which have been shown in the legend.

6. In method, will need to provide more details how to do multivariate adjusted models that are shown in Table 3.

Response: Thank you for the suggestions. We provided more details in method part referring to multivariate adjusted models.

7. Clinical implications of this study and future directions are needed to improve

Response: Thank you for this comment. We added this in conclusions section.

Reviewer 2 Report

The manuscript by Du, W et. al. entitled “Thirty-year urbanization trajectories and obesity in modernizing China” is a study with an aim to determine the impact of long-term (18 years) urbanization changes on obesity outcomes among Chinese population.

The manuscript is little bit indistinctly written so some parts are hard to understand what the authors wanted to point out.

Some technical comments:

English proofreading is suggested there are some unusual sentence constructions in the manuscript.

Why OV+ (what is the purpose of the plus, it is generally known what the overweight stands for; this way you can mislead the reader)

Row 47 – word is missing (…in rural and urban…what?? Areas or?)

Row 108 – definition of overweight means above 24,9, and obesity above 29,9 kg/m2 – it is not recommended but if you use some Chinese definition of OV/OB literature reference is needed

Row 119 – MET -one metabolic equivalent – why * ??

Row 139-141 – it should not be a part of the manuscript

Figure 1 is unclear for the readers

Where are data for 2018 in table 1?? It would be much more interesting to see the average BMI by each year of research and not only share of respondents

Fig 2 – why predicted BMI when you have actual BMI or if you want to put predicted than you need to compare it with actual and show difference between model and the actual situation

Why you didn’t try to compare your results with more similar obtained worldwide. Also, lot of results wasn’t discussed in discussion part.

Some questions which should be discussed:

  • How we can compare subjects with “at least two survey visits” e.g. 2015 and 2018 vs. 2009 and 2018 – I assume very different results can be obtained especially due globalization impact

Author Response

Thank you for the invaluable comments that have enabled us to improve this manuscript. The manuscript has been revised according to the comments and journal style. Below is a detailed response:

1. English proofreading is suggested there are some unusual sentence constructions in the manuscript.

Response: Thank you for the suggestion. English language in this manuscript has been checked using mdpi editing services.

2. Why OV+ (what is the purpose of the plus, it is generally known what the overweight stands for; this way you can mislead the reader)

Response: Thank you for this comment. We used OV+ here to represent combined overweight and obesity, with BMI≥24kg/m2, because OV+ could indicate more evidence in relation to increased BMI. To avoid confusion, we changed OV+ to OO (overweight+obesity) in the revised manuscript.

3. Row 47 – word is missing (…in rural and urban…what?? Areas or?)

Response: Thank you very much for this comment. We added “areas” in this sentence.

4. Row 108 – definition of overweight means above 24,9, and obesity above 29,9 kg/m2 – it is not recommended but if you use some Chinese definition of OV/OB literature reference is needed

Response: Thank you for the comments. We added the reference on Chinese definition of OV/OB.

5. Row 119 – MET -one metabolic equivalent – why * ??

Response: Thank you for this comment. We meant “the metabolic equivalent of task hours per week”, and revised this in the manuscript to avoid confusion.

6. Row 139-141 – it should not be a part of the manuscript

Response: Thanks for this comment. We deleted this paragraph. 

7. Figure 1 is unclear for the readers

Response: Thank you for this comment. We added the figure title, x- and y-axis to make it more clear.

8. Where are data for 2018 in table 1?? It would be much more interesting to see the average BMI by each year of research and not only share of respondents

Response: Thank you for the comments. Table 1 described the characteristics of baseline sample, who were followed at least two waves. So there would not be data for 2018 in table 1. The share of respondents in table 1 illustrated distribution of the sample enrolled time. We reported the BMI by each year of research in Figure 2.

9. Fig 2 – why predicted BMI when you have actual BMI or if you want to put predicted than you need to compare it with actual and show difference between model and the actual situation

Response: Thank you for the suggestion. Figure 2 aimed to interpret our model-based findings. The predicted BMI was calculated at given combination of urbanization trajectory and study year while holding the covariates at their mean values.

10. Why you didn’t try to compare your results with more similar obtained worldwide. Also, lot of results wasn’t discussed in discussion part.

Response: Thank you for these valuable comments. We added references in discussion part on comparisons with other regions and potential explanations. Although there were limited references on urbanicity trajectories and obesity, the authors try to discuss the results not point-to-point, but in several dimensions, including our main findings, comparisons with previous studies (regions, social-demography, time stages, gender disparites)  and related explanations, as well as strengths and limitations.  

11. Some questions which should be discussed:

How we can compare subjects with “at least two survey visits” e.g. 2015 and 2018 vs. 2009 and 2018 – I assume very different results can be obtained especially due globalization impact

Response: Thank you for this comment. Yes, time is a very important covariate in exploring the hypothesis. So that, we include survey year, as well as baseline age and follow-up age in the adjusted models, to control potential globalization impact. 

Reviewer 3 Report

This study is a very important topic in modern public health. However, the manuscript needs to be revised in order to assess the validity of the study. In particular, the definitions of overweight and obesity needs to be described more precisely.

1) Provide evidence (references, etc.) for the cutoff both of overweight+ (BMI≥24) and obesity+ (BMI≥28).

2) Provide evidence (references, etc.) for the three age categories.

3) Add an explanation of the units for the vertical and horizontal axes in Figure 1.

4) Clearly explain how this study measured the height and weight (i.e., measured with the same equipment or not).

5) Make sure the references in Table 2 are correct (Isn't it 1 ?).

6) After revising the above points, please show the STROBE guidelines (https://www.strobe-statement.org/checklists/).

Author Response

Thank you for the invaluable comments that have enabled us to improve this manuscript. The manuscript has been revised according to the comments and journal style. Below is a detailed response:

This study is a very important topic in modern public health. However, the manuscript needs to be revised in order to assess the validity of the study. In particular, the definitions of overweight and obesity needs to be described more precisely.

Response: Thank you for the comments.

1. Provide evidence (references, etc.) for the cutoff both of overweight+ (BMI≥24) and obesity+ (BMI≥28).

Response: Thank you for the suggestion. We added the reference on Chinese definition of OO/OB.

2. Provide evidence (references, etc.) for the three age categories.

Response: Thank you for this comment. We divided the three age categories according to our analysis purpose and previous research. Reference was added here.

3. Add an explanation of the units for the vertical and horizontal axes in Figure 1.

Response: Thank you for the suggestion. We added x- and y- axis in figure1.

4. Clearly explain how this study measured the height and weight (i.e., measured with the same equipment or not).

Response: Thank you for the suggestion. We added the protocols in field as well as equipment for measured height and weight.

5. Make sure the references in Table 2 are correct (Isn't it 1 ?).

Response: Thank you for this comment. We checked table2 and found no mistakes.

6. After revising the above points, please show the STROBE guidelines (https://www.strobe-statement.org/checklists/).

Response: Thank you for this comment. We completed the STROBE guidelines table according to revised manuscript. Please see the attachment. 

Round 2

Reviewer 2 Report

I would like to thank the authors for their efforts to improve the manuscript.

Some rearrangements have been made to the text. The paper is now somewhat clearer.

Figure 2 is still little bit unreadable – the lines are too thick so differences is hard to see. Please remove „predicted“ because it is not predicted but measured weight.

Author Response

1.I would like to thank the authors for their efforts to improve the manuscript. Some rearrangements have been made to the text. The paper is now somewhat clearer.

Response: Thank you for your comments. 

2.Figure 2 is still little bit unreadable – the lines are too thick so differences is hard to see. Please remove „predicted“ because it is not predicted but measured weight.

Response: Thank you for the suggestion. We modified the lines in figure 2 to make it easy to see the differences. 

Sorry for our unclear explanation in previous letter. The weight values showed  in figure 2 were estimated via mixed model using measured weight. So that, we did not remove "predicted“ here. 

Reviewer 3 Report

This manuscript has been properly revised.

I have no additional comments.

Author Response

This manuscript has been properly revised.

I have no additional comments.

Response: Thank you again for your valuable comments.